# FreeMask: Synthetic Images with Dense Annotations Make Stronger Segmentation Models

**Lihe Yang**[1]    **Xiaogang Xu**[2,3]    **Bingyi Kang**[4]    **Yinghuan Shi**[5]    **Hengshuang Zhao**[1]*

[1]The University of Hong Kong    [2]Zhejiang Lab    [3]Zhejiang University
[4]ByteDance    [5]Nanjing University

https://github.com/LiheYoung/FreeMask

## Abstract

Semantic segmentation has witnessed tremendous progress due to the proposal of various advanced network architectures. However, they are extremely hungry for delicate annotations to train, and the acquisition is laborious and unaffordable. Therefore, we present `FreeMask` in this work, which resorts to synthetic images from generative models to ease the burden of both data collection and annotation procedures. Concretely, we first synthesize abundant training images conditioned on the semantic masks provided by realistic datasets. This yields extra well-aligned image-mask training pairs for semantic segmentation models. We surprisingly observe that, *solely* trained with synthetic images, we already achieve comparable performance with real ones (*e.g.*, 48.3 *vs.* 48.5 mIoU on ADE20K, and 49.3 *vs.* 50.5 on COCO-Stuff). Then, we investigate the role of synthetic images by joint training with real images, or pre-training for real images. Meantime, we design a robust filtering principle to suppress incorrectly synthesized regions. In addition, we propose to inequally treat different semantic masks to prioritize those harder ones and sample more corresponding synthetic images for them. As a result, either jointly trained or pre-trained with our filtered and re-sampled synthesized images, segmentation models can be greatly enhanced, *e.g.*, 48.7 $\rightarrow$ 52.0 on ADE20K.

## 1  Introduction

Semantic segmentation aims to provide pixel-wise dense predictions for a whole image. With advanced network architectures, the performance has been steadily boosted over the years, and has been applied to various real-world applications, *e.g.*, autonomous driving [40, 3], semantic editing [39, 56], scene reconstruction [36], *etc*. However, both academic benchmarks and realistic scenarios, suffer from the laborious labeling process and unaffordable cost, leading to limited training pairs, *e.g.*, Cityscapes [16] with only 3K labeled images and ADE20K [76] with only 20K labeled images. We are wondering, whether these modern architectures can be further harvested when more synthetic training data from the corresponding scenes are fed.

Fortunately, the research of generative models has reached certain milestones. Especially, denoising diffusion probabilistic models (DDPMs) [27, 51], are achieving astonishing text-to-image synthesis ability in the recent three years. *Can these synthetic images from generative models benefit our semantic segmentation task?* There have been some concurrent works [25, 54, 4, 2, 64] sharing similar motivations as ours, *i.e.*, using synthetic data to enhance the semantic understanding tasks. However, they have only explored the classification task, which is relatively easier to be boosted even with raw unlabeled images [11, 24, 12, 23]. Moreover, they mostly focus on the few-shot scenario, rather than taking the challenging fully-supervised scenario into account. In comparison, in this work,

---

*Corresponding author

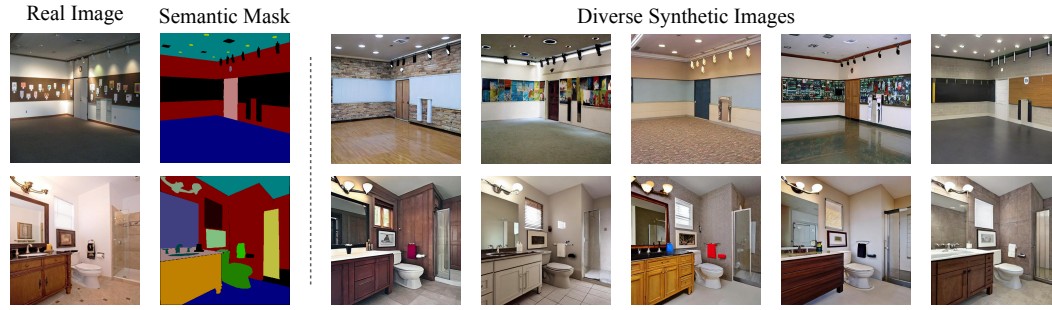

Figure 1: Generated synthetic images conditioned on semantic masks. These synthetic images are high-quality and diverse. We wish to utilize these extra training pairs (synthetic image and its conditioned mask) to boost the fully-supervised baseline, which is trained only with real images.

we wish to leverage synthetic images to directly improve the *fully-supervised* performance of the fundamental dense prediction task–*semantic segmentation*.

To formulate credible and controllable image synthesis to boost semantic segmentation, we propose to employ the SOTA diffusion models with semantic masks as conditional inputs. Unconditional image synthesis requires manual annotations [73, 37], or will suffer from the issue of misalignment between synthetic images and the corresponding dense annotations [20]. Especially, we adopt an off-the-shelf semantic image synthesis model FreestyleNet [71] to perform mask-to-image synthesis.

However, even with such a conditional generator, there are still several challenges: (1) The synthesized image may be of a different domain from our target images. It is necessary to generate in-domain images. (2) The synthesis results cannot be perfect. Some failure cases will deteriorate the subsequent learning process. (3) The performance of our targeted fully-supervised baseline has almost been saturated with optimal hyper-parameters. Thus, the synthesis should be emphasized on the hard samples that are crucial for improving the generalization performance.

To solve challenge (1), we first prompt the output space of FreestyleNet [71] to be closer to the target datasets, *i.e.*, FreestyleNet is pre-trained with a powerful Stable Diffusion model [51], and then fine-tuned on specific mask-to-image synthesis datasets, *e.g.*, ADE20K [76] and COCO-Stuff [9]. Hence, it not only acquires generic knowledge from the large-scale pre-training set of Stable Diffusion, but also is capable of yielding in-domain images conditioned on the semantic masks. As depicted in Figure 1, we can produce diverse high-fidelity synthetic images given existing training masks from our targeted datasets. Then, each synthetic image is paired with its conditioned semantic mask, forming a new training sample with dense annotations.

However, when training models with these synthetic pairs, we observe that it is evidently inferior to the counterpart of training with real images, since there are still some image regions synthesized unsatisfactorily. These poorly synthesized images will have a negative impact on training, even overriding the positive influence brought by the large dataset size, leading to the challenge (2). To this end, we propose to filter synthetic images in a self-adaptive manner at the pixel level according to a class-level evaluation. Intuitively, incorrectly synthesized regions or images will exhibit significant losses, if evaluated under a model pre-trained on real images. Motivated by this, we first obtain the average loss of each semantic class, which is calculated through the whole synthetic set with a real-image pre-trained model. Then we suppress noisy synthetic regions under this principle: if the loss of a synthetic pixel surpasses the average loss of its corresponding class by a certain margin, it will be marked as a noisy pixel and ignored during loss computation. Despite its simplicity, this strategy significantly improves the model performance when trained with synthetic images.

Furthermore, we implement hardness-aware data synthesis to generate more hard samples, effectively solving challenge (3) and prompting the training better. We observe that different semantic masks are of varying hardness. For example, the mask in the first row of Figure 1, which exhibits regular object structures and limited semantic classes, is obviously much easier to learn than the mask in the second row. These over-easy masks and their corresponding synthetic images will dominate the learning process as well as the gradient. Therefore, we propose to inequally treat these semantic masks during conditional image synthesis. Concretely, with recorded class-wise average losses, we can calculate the overall loss of a semantic mask, which can represent its global hardness. For a semantic mask, we then determine the number of its required synthetic images based on its hardness. The harder it

is, the more synthetic images will be generated for it. Consequently, the model can be trained more sufficiently on these challenging layouts. Our contributions lie in four aspects:

- We present a new roadmap to enhance *fully-supervised* semantic segmentation via generating *densely annotated* synthetic images with generative models. Our data-centric perspective is orthogonal to the widely explored model-centric (*e.g.*, network architecture) perspective.

- We propose a robust filtering criterion to suppress noisy synthetic samples at the pixel and class levels. Moreover, we design an effective metric to indicate the hardness of semantic masks, and propose to discriminatively sample more synthetic images for harder masks.

- With our filtering criterion and sampling principle, the model trained solely with synthetic images can achieve comparable performance with the counterpart of real images, *e.g.*, 48.3 *vs*. 48.5 mIoU on ADE20K and 49.3 *vs*. 50.5 on COCO-Stuff.

- We investigate different paradigms to leverage synthetic images, including (1) jointly training on real and synthetic images, (2) pre-training on synthetic ones and then fine-tuning with real ones. We observe remarkable gains (*e.g.*, 48.7 → 52.0) under both paradigms.

## 2 Related Work

**Semantic segmentation.** Earlier semantic segmentation works focus on designing various attention mechanisms [68, 29], enlarging receptive fields [74, 10, 44], and extracting finer-grained information [38]. With the introduction of the Transformer architecture [21] to CV, advanced semantic segmentation models [70, 75, 14] are mostly Transformer-based. Among them, MaskFormer [15] presents a new perspective to perform semantic segmentation via mask classification. Orthogonal to these model-centric researches, our work especially highlights the important role of training data.

**Generative models.** Generative models have been developed for several years to synthesize photo-realistic images and can be divided into two main categories, *i.e.*, unconditional and conditional models. The input of unconditional generative models is only noise, and the conditional networks take inputs of multi-modality data (*e.g.*, labels, text, layouts) and require the outputs to be consistent with the conditional inputs. The structures of generative models have evolved rapidly in recent years. Generative Adversarial Networks (GANs) [22], Variational AutoEncoder (VAE) [35], and Diffusion models [58] are the most representative ones among them. GANs are formulated with the theoretical basis of Nash equilibrium, where the generator is trained with a set discriminator. Several excellent GANs have been proposed in recent years to synthesize high-quality photos, *e.g.*, BigGAN [8] and StyleGAN [32, 33, 31, 55]. VAEs belong to the families of variational Bayesian methods and are optimized with an Evidence lower bound (ELBO). VQ-VAE [48] is one famous model which can generate high-resolution images effectively. Diffusion models [58, 28, 42] have recently emerged as a class of promising and effective generative models. It first shows better synthesis ability than GANs in 2021 [19], and then various diffusion models have been designed and achieved outstanding results, *e.g.*, Stable Diffusion [51], Dalle [49, 46], Imagen [53], *etc*. Moreover, GANs [43, 63] and Diffusion models [72, 67, 71] can both complete image synthesis from semantic layouts.

**Learning from synthetic images.** There are three main sources of synthetic images. (1) From computer graphics engines [50, 52]. These synthetic images with dense annotations are widely adopted in domain generalization [65] and unsupervised domain adaptation [17]. However, there exists a large domain gap between them and real images, making them struggle to benefit the fully-supervised baseline. (2) From procedural image programs. These inspiring works [7, 34, 6, 62] delicately design abundant mathematical formulas to sample synthetic images. Despite promising results obtained in ImageNet fine-tuning, it is laborious to construct effective formulas, and they are still even lagging behind the basic ImageNet pre-training in dense prediction tasks. (3) From generative models. Our work belongs to this research line. We provide a detailed review below.

Early works [1, 59, 73, 37, 30] have utilized GANs to synthesize extra training samples. Recently, there have been some initial attempts to apply Diffusion models [51] to the classification task. For example, He *et al*. [25] examined the effectiveness of synthetic images in data-scarce settings (zero-shot and few-shot). The synthetic images are also found to serve as a promising pre-training data source for downstream tasks. In addition, Sariyildiz *et al*. [54] manages to replace the whole ImageNet images [18] with pure synthetic ones. However, it still reports a huge performance gap compared with its real counterpart. Different from previous works that directly produce non-existing

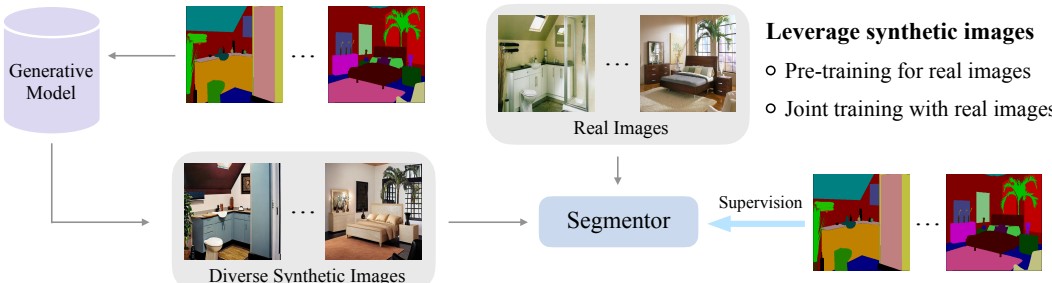

Figure 2: Illustration of our proposed roadmap to boost fully-supervised semantic segmentation with densely annotated synthetic images. The generative model can synthesize diverse new images conditioned on the semantic mask from target datasets. Synthetic images and conditioned masks form new training pairs. We investigate pre-training and joint training to leverage these synthetic images.

images, Trabucco *et al.* [64] proposes to leverage Stable Diffusion [51] to edit existing training images, *e.g.*, altering the color. The most latest work [2] shares a similar motivation as us in that it also wishes to enhance the fully-supervised performance. Nevertheless, it only addresses the classification task, which is relatively easier to be boosted even directly with raw unlabeled images [11, 24, 23] or pre-training image-text pairs [45]. Distinguished from it, we aim at the challenging semantic segmentation task which requires more precise annotations than image-text pairs. More importantly, we further propose two strategies to learn from synthetic images more robustly and discriminatively. We also discuss two learning paradigms.

**Discussion with semi-supervised learning (SSL).** Our strategy is different from SSL, *i.e.*, we synthesize new images from masks, while SSL predicts pseudo masks for extra unlabeled images. Our proposed roadmap is better than SSL in three aspects: (1) It is not trivial to ensure the extra unlabeled images for SSL are of the same domain as labeled images, while our target-set fine-tuned generative model can consistently yield in-domain images. (2) It is much cheaper and time-saving to synthesize abundant and almost unlimited images from semantic masks, compared with manually collecting appropriate unlabeled images. (3) We observe it is more precise in structures to synthesize images from masks than predict masks for images, especially in boundary regions.

## 3 Method

In this section, we first introduce our synthesis pipeline in Section 3.1. Then, we clarify our filtering criterion for noisy synthetic regions in Section 3.2. Following this, we further present a hardness-aware re-sampling principle to emphasize hard-to-learn semantic masks in Section 3.3. Lastly, we discuss two paradigms to learn from these high-quality synthetic images in Section 3.4.

### 3.1 Synthesizing Densely Annotated Images

Semantic image synthesis [66, 61, 63] aims to generate a photo-realistic image conditioned on a given semantic layout. Among these approaches, recent works [67, 71] have shown that large-scale text-to-image diffusion models can be trained to synthesize high-quality and diverse results. Motivated by this, we adopt the most recent work FreestyleNet [71], to synthesize additional training images conditioned on the training masks provided by our targeted datasets. Take ADE20K [76] as an example, it only contains 20K image-mask training pairs. However, aided by the mask-to-image synthesis network, we can construct almost unlimited synthetic images from these 20K semantic masks. Conditioned on each semantic mask, if we synthesize $K$ images, we will obtain an artificial training set whose scale is $K\times$ larger than the original real training set. After synthesis, each synthetic image can be paired with the conditioned mask to form a new training sample with dense annotations. We provide an illustration of our pipeline in Figure 2.

**Discussion.** We construct densely annotated synthetic images through mask-to-image synthesis. However, some works adopt a different or even "reversed" approach. We briefly discuss them and make a comparison. DatasetGAN [73] and BigDatasetGAN [37] first produce synthetic images in an unconditional manner. Their dense annotations will be assigned by human annotators. Then the model will learn from these training pairs and predict pseudo masks for new synthetic images. The

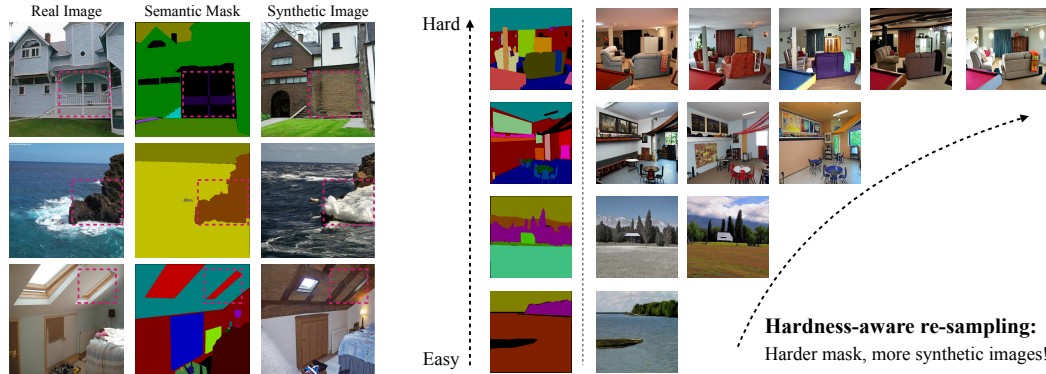

Figure 3: Visualization of failure cases during synthesis. Incorrect synthetic regions are highlighted with pink boxes.

Figure 4: We propose a mask-level hardness-aware re-sampling strategy for synthesis. Easy masks are of regular shapes and cover limited and common semantics, while hard masks are the opposite.

main drawback of such methods is the involvement of expensive human efforts. E.g., 5K manual annotations are needed to train classification tasks on ImageNet, which is costly, let alone providing the annotation of dense semantic masks.

To avoid the cost of manual annotations, a recent work [20] directly trains a segmentation model on real pairs and then uses it to produce pseudo masks for synthetic images. It is very close to the semi-supervised scenario (only replacing real unlabeled images with synthetic unlabeled images). However, since there is a domain gap between the real and synthetic images inevitably, we cannot guarantee the correctness of the pseudo masks.

On the other hand, our mask-conditioned image synthesis can avoid the issue of annotation cost and meanwhile produce accurately aligned image-mask pairs, especially in boundary regions. Moreover, our adopted conditional image synthesis is more controllable than unconditioned synthesis. For example, we can better control the class distributions of synthetic images by editing the semantic masks. This has the potential to help alleviate the long-tail distributions. Furthermore, with mask-to-image synthesis, we can synthesize more hard-to-learn scenes (defined by semantic masks) for models to learn, which will be discussed in Section 3.3.

### 3.2 Filtering Noisy Synthetic Regions

**Motivation.** As illustrated in Figure 3, although visually appealing, there still exist some artifacts in synthetic images. Blindly using all synthetic samples for training will cause the model to be severely deteriorated by these noisy regions, overshadowing the performance gain brought by additional synthetic images, especially when the synthetic set is large. This will be demonstrated in Section 4: directly scaling up the number of *raw* synthetic images for training cannot progressively yield better performance on the real validation set, even achieving worse results.

**Our filtering strategy.** To this end, we propose to adaptively filter harmful synthetic regions via computing hardness score at the class level and conducting filter at the pixel level. Specifically, detrimental synthesized regions denote those regions that do not match corresponding semantic masks well. They usually exhibit large losses if evaluated with a well-trained semantic segmentation model. Motivated by this, we filter synthetic pixels based on the following criterion: if the loss of a synthetic pixel belonging to semantic label $j$ has surpassed the recorded average loss of all class-$j$ pixels by a certain margin, this synthetic pixel will be marked as a noisy one and simply ignored during loss computation. To instantiate this idea, we use a semantic segmentation model which is pre-trained on real images to calculate dense loss maps $\{\mathbf{S}^i\}_{i=1}^N$ ($\mathbf{S}_i = \{s_k, k \in [1, H \times W]\}$ where $s_k$ is the pixel-wise loss value, $H$ and $W$ denote the map height and width, respectively) for all $N$ synthetic images with their semantic masks $\{\mathbf{M}^i\}_{i=1}^N$. Then, we can obtain the average loss $h_j$ of class $j$ by:

$$h_j = \sum_{i=1}^N \sum_{hw}^{HW} [\mathbb{1}(\mathbf{M}_{hw}^i = j) \times \mathbf{S}_{hw}^i] / \sum_{i=1}^N \sum_{hw}^{HW} \mathbb{1}(\mathbf{M}_{hw}^i = j), \qquad (1)$$

where $\mathbb{1}(x) = 1$ if $x$ is True, otherwise 0, and $hw$ indicates the pixel location of $(h, w)$. The $h_j$ indeed reflects the learning hardness of semantic class $j$ in the corresponding target dataset, *i.e.*,

harder classes tend to have larger $h$. We will further reuse $h$ as an effective indicator for mask-level hardness in Section 3.3.

With $K$ average losses of $K$ semantic classes, we can filter noisy synthetic pixels. For a synthetic pixel $k$ with label $j$, if its loss $s_k > h_j \cdot \alpha$, it will be deemed as a potentially noisy pixel, where $\alpha$ here serves as a tolerance margin. The smaller $\alpha$ is, the more synthetic pixels will be filtered and ignored during loss computation for better safety. Meanwhile, too small $\alpha$ will also lead to the remaining pixels not being informative enough for our model to learn. Fortunately, our filtering strategy is robust to $\alpha$, as the performance is stable under different $\alpha$ ranging from 1 to 2.

**Effects.** Despite the simplicity of this filtering criterion, it significantly boosts the generalization performance of the target model when trained with synthetic images and tested in real images. It is even on par with the performance achieved by real training images (less than 1% margin in test mIoU). This demonstrates its great potential in replacing real images with synthetic images in the future, especially when dealing with some privacy-sensitive application scenarios.

### 3.3 Re-sampling Synthetic Images based on Mask-level Hardness

**Motivation.** We further observe that some semantic masks are easily learnable layouts, while some masks depict relatively messy scenes that are harder to learn, as shown in Figure 4. These simple layouts exhibit small overall losses even if a never encountered synthetic image is fed. Thus, the benefit obtained by learning from such layouts is usually trivial. On the other hand, those hard-to-learn scenes require more attention during image synthesis and subsequent training.

**Our re-sampling strategy.** To address this issue, we present a simple and straightforward principle to quantify the global hardness of each semantic mask. Based on the mask-level hardness, we further propose to discriminatively sample synthetic images from semantic masks, *i.e.*, prioritizing harder masks. Concretely, to obtain the hardness, we reuse the class-wise average loss $h$ defined in Equation 1. A minor difference is that $h$ here is calculated on the real training pairs to more precisely reflect the difficulty of training masks. The global mask-level hardness can be viewed as an aggregation across the hardness of all its pixels, and the hardness of each pixel is associated with its semantic label. Formally, the hardness $p_i$ of the $i$-th mask can be computed by $p_i = \sum_{hw}^{HW} h_{\mathbf{M}_{hw}^i}$, where $\mathbf{M}_{hw}^i$ represents the class index of position $(h, w)$ in the semantic mask map.

Following this, we sort totally $N$ semantic masks according to their quantitative hardness. We can then sample more synthetic images for harder masks based on their rankings among all masks. More specifically, for the $p$-largest-hardness mask, the number of its synthetic images $n_p$ is determined by:

$$n_p = N_{\max} \cdot (N - p)/N, \tag{2}$$

where $N_{\max}$ is our pre-defined maximum number of synthetic images for a single mask. The $p$ ranges from 0 to $(N - 1)$, and $n_p$ will be rounded up to an integer, ranging from 1 to $N_{\max}$.

**Effects.** Our hardness-aware synthesis strategy can avoid the learning process being dominated by over-easy image-mask pairs and avoid vanishing gradients. It also enables the challenging layouts to be sufficiently learned and thus achieves better performance. Indeed, to some extent, it shares a similar spirit with online hard example mining (OHEM) [57]. Meanwhile, it fundamentally differs from OHEM in that we synthesize more novel hard samples for the model to learn, rather than just implicitly enlarging the importance of existing hard samples as OHEM does.

### 3.4 Learning from Synthetic Images

We discuss two paradigms to use the synthetic training set for the learning of semantic segmentation.

**Pre-training.** The segmentation model is first trained with densely annotated synthetic images for the same number of iterations as real images. During synthetic pre-training, we test the transferring performance on the *real* validation set. By comparing this performance with that obtained by real training images, we can gain better intuition on the gap between synthetic and real images. Next, we further fine-tune the pre-trained model with real images. This paradigm can effectively overcome the noise caused by synthetic images, especially when synthetic images are of relatively poor quality.

**Joint training.** The synthetic set is usually larger than the real set. Meanwhile, the real training pairs are of higher quality than synthetic ones. Therefore, we first over-sample real images to the

| Model | Training Data | | mIoU |
|-------|-----|-----|------|
| | Real | Synthetic | |
| SegFormer-B4 [70] | ✓ | | 48.5 |
| | | ✓ | 48.3 |
| Mask2Former [14] | ✓ | | 56.0 |
| | | ✓ | 53.5 |

Table 1: Transferring performance on **ADE20K**. The validation set is real images. The Mask2Former uses a Swin-L [41] backbone pre-trained on ImageNet-22K.

| Model | Training Data | | mIoU |
|-------|-----|-----|------|
| | Real | Synthetic | |
| SegFormer-B4 [70] | ✓ | | 45.8 |
| | | ✓ | 44.4 |
| ViT-Adapter [13] | ✓ | | 50.5 |
| | | ✓ | 49.3 |

Table 2: Transferring performance on **COCO-Stuff**. The validation set is real images. The ViT-Adapter uses UperNet [69] and a BEiT-L backbone [5].

| Model | Backbone | Training Data | | mIoU | Δ |
|-------|----------|------|-----|------|---|
| | | Real | Synthetic | | |
| Mask2Former [14] | Swin-T [41] | ✓ | | 48.7 | ↑ **3.3%** |
| | | ✓ | ✓ | **52.0** | |
| | Swin-S [41] | ✓ | | 51.6 | ↑ **1.7%** |
| | | ✓ | ✓ | **53.3** | |
| | Swin-B [41] | ✓ | | 52.4 | ↑ **1.3%** |
| | | ✓ | ✓ | **53.7** | |
| SegFormer [70] | MiT-B2 [70] | ✓ | | 45.6 | ↑ **2.3%** |
| | | ✓ | ✓ | **47.9** | |
| | MiT-B4 [70] | ✓ | | 48.5 | ↑ **2.1%** |
| | | ✓ | ✓ | **50.6** | |
| Segmenter [60] | ViT-S [21] | ✓ | | 46.2 | ↑ **1.7%** |
| | | ✓ | ✓ | **47.9** | |
| | ViT-B [21] | ✓ | | 49.6 | ↑ **1.5%** |
| | | ✓ | ✓ | **51.1** | |

Table 3: Integrating synthetic images by **joint training** on **ADE20K** to enhance the real-image baseline.

same number as synthetic images. Then we simply train a segmentation model on the combined set. Compared with pre-training, joint training can better preserve the knowledge acquired from the larger-scale synthetic set. It may be more promising when the transferring performance of synthetic images is exceptionally close to that of real images.

# 4 Experiment

We describe the implementation details in Section 4.1. Then we report the transferring performance of synthetic training pairs to real validation set in Section 4.2. Further, we leverage synthetic images to boost the fully-supervised baseline in Section 4.3. Finally, we conduct ablation studies in Section 4.4.

## 4.1 Implementation Details

We evaluate the effectiveness of synthetic images on two widely adopted semantic segmentation benchmarks, *i.e.*, ADE20K [76] and COCO-Stuff [9]. They are highly challenging due to the complex taxonomy. COCO-Stuff is composed of 118,287/5,000 training/validation images, spanning over 171 semantic classes. In comparison, ADE20K is more limited in training images, containing 20,210/2,000 training/validation images and covering 150 classes.

We use two open-source checkpoints from FreestyleNet [71] for ADE20K and COCO respectively, to produce photo-realistic synthetic images conditioned on training masks. We synthesize at most 20 extra images for each semantic mask on ADE20K, *i.e.*, $N_{max} = 20$ in Equation 2. On COCO, considering its large number of semantic masks, we set a smaller $N_{max} = 6$ due to the limitation of disk space on our server. By default, we set the tolerance margin $\alpha$ as 1.25 for all experiments. In pre-training, we exactly follow the hyper-parameters of regular training. In joint training, we

| Model | Backbone | Training Data | | mIoU | Δ |
|---|---|---|---|---|---|
| | | Real | Synthetic | | |
| Mask2Former [14] | Swin-T [41] | ✓ | | 44.5 | |
| | | ✓ | ✓ | **46.4** | ↑**1.9%** |
| | Swin-S [41] | ✓ | | 46.8 | |
| | | ✓ | ✓ | **47.6** | ↑**0.8%** |
| SegFormer [70] | MiT-B2 [70] | ✓ | | 43.5 | |
| | | ✓ | ✓ | **44.2** | ↑**0.7%** |
| | MiT-B4 [70] | ✓ | | 45.8 | |
| | | ✓ | ✓ | **46.6** | ↑**0.8%** |
| Segmenter [60] | ViT-S [21] | ✓ | | 43.5 | |
| | | ✓ | ✓ | **44.8** | ↑**1.3%** |
| | ViT-B [21] | ✓ | | 46.0 | |
| | | ✓ | ✓ | **47.5** | ↑**1.5%** |

Table 4: Integrating synthetic images by **joint training** on **COCO-Stuff** to enhance the real-image baseline.

| Model | Backbone | Training Data | | mIoU | Δ |
|---|---|---|---|---|---|
| | | Real: Fine-tune | Synthetic: Pre-train | | |
| SegFormer [70] | MiT-B2 [70] | ✓ | | 45.6 | |
| | | ✓ | ✓ | **47.0** | ↑**1.4%** |
| | MiT-B4 [70] | ✓ | | 48.5 | |
| | | ✓ | ✓ | **50.6** | ↑**2.1%** |
| Segmenter [60] | ViT-S [21] | ✓ | | 46.2 | |
| | | ✓ | ✓ | **47.2** | ↑**1.0%** |
| | ViT-B [21] | ✓ | | 49.6 | |
| | | ✓ | ✓ | **50.6** | ↑**1.0%** |
| Mask2Former [14] | Swin-L[†] [41] | ✓ | | 56.0 | |
| | | ✓ | ✓ | **56.4** | ↑**0.4%** |

Table 5: Integrating synthetic images by **pre-training** on **ADE20K** to enhance the real-image baseline. †: The Swin-L backbone is pre-trained on the large-scale ImageNet-22K dataset.

over-sample real images to the same number of synthetic images . The learning rate and batch size are the same as the regular training paradigm. Due to the actually halved batch size of real images in each iteration, we double the training iterations to iterate over real training images for the same epochs as regular training. Other hyper-parameters are detailed in the appendix.

## 4.2 Transferring from Synthetic Training Images to Real Test Images

Existing semantic image synthesis works [61, 71] examine the synthesis ability by computing the FID score [26]. However, we observe this metric is not well aligned with the benefits brought by augmenting real image datasets with synthetic images, as also suggested in [47]. Therefore, similar to the Classification Accuracy Score proposed in [47], we exhibit the synthesis quality by transferring the model trained on synthetic images to real test images. The semantic segmentation score (mean Intersection-over-Union) on real test images is a robust indicator for synthesized images.

Using this score, we first validate the quality of our filtered and re-sampled synthetic images. We train a model using densely annotated synthetic images alone, and compare it with the counterpart of only using real training images. As shown in Table 1 of ADE20K, under the widely adopted SegFormer [70] model with a MiT-B4 backbone, the performance gap between using synthetic training images and real training images is negligible, *i.e.*, 48.3% *vs*. 48.5% mIoU (-0.2%). Then we attempt one of the most powerful semantic segmentation models, *i.e.*, Mask2Former [14] with a heavy Swin-L [41] backbone pre-trained on the large-scale ImageNet-22K. Although the real-data trained model achieves a fantastic result (56.0%), our synthetic training images can also yield a strong

| Dataset | Our Strategies | | mIoU | Dataset | Our Strategies | | mIoU |
|---|---|---|---|---|---|---|---|
| | Filtering | Re-sampling | | | Filtering | Re-sampling | |
| ADE20K | | | 43.3 | COCO | | | 48.0 |
| | ✓ | | 47.4 | | ✓ | | 48.9 |
| (48.5) | | ✓ | 45.1 | (50.5) | | ✓ | 48.6 |
| | ✓ | ✓ | **48.3** | | ✓ | ✓ | **49.3** |

Table 6: Effectiveness of our filtering and re-sampling strategies for **transferring performance**. The results below the dataset names denote the transferring performance obtained with real training images.

| Our Strategies | | mIoU |
|---|---|---|
| Filtering | Re-sampling | |
| | | 48.2 |
| ✓ | | 49.9 |
| | ✓ | 49.3 |
| ✓ | ✓ | **50.6** |

Table 7: Effectiveness of our strategies when **jointly training** two sources of synthetic and real images.

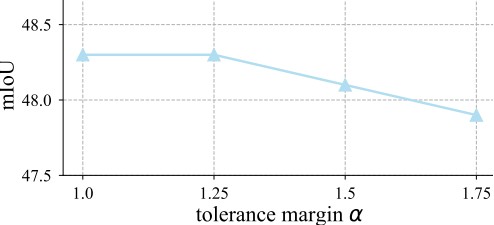

Figure 5: Ablation study on the hyper-parameter $\alpha$.

model (53.5%). The marginal performance gap (-2.5%) is impressive, especially considering the large gap (around -8%) in the latest work [2] for synthetic classification.

The corresponding experimental results on COCO are shown in Table 2, and the observation is consistent. Either with SegFormer [70] or the state-of-the-art model ViT-Adapter [13], the transferring performance gap between synthetic and real images is constrained within -1.5%.

We have demonstrated the impressive transferring ability of our filtered and re-sampled synthetic training images toward real test images. It indicates they have the great potential to completely replace the real training images in the future. They can play a significantly important role in many privacy-sensitive application scenarios. Furthermore, in Section 4.4, we will show that the transferring performance will decrease a lot without our filtering and re-sampling strategies.

## 4.3 Enhancing Fully-supervised Baseline with Synthetic Images

We further take full advantage of these synthetic images to enhance the fully-supervised baseline, which is merely trained with real images before. We investigate two training paradigms to integrate synthetic and real training images, *i.e.*, (1) jointly training with both real and synthetic images, and (2) pre-training with synthetic ones, then fine-tuning with higher-quality real ones.

Under the joint training paradigm, on ADE20K of Table 3, with extra synthetic images, we examine the performance of a wide range of modern architectures. All models are evidently boosted by larger than 1%. Among them, the widely adopted Mask2Former-Swin-T is improved from 48.7% → 52.0% mIoU (+3.3%), and SegFormer-B4 is boosted from 48.5% → 50.6% (+2.1%). Also, as demonstrated in Table 4 for COCO-Stuff, although the number of real training images is already large enough (120K), our densely annotated synthetic images further push the upper bound of existing state-of-the-art models. The Mask2Former-Swin-T model is non-trivially enhanced by 1.9% mIoU from 44.5% → 46.4%.

Moreover, under the pre-training paradigm, we report results on ADE20K in Table 5. The SegFormer-B4 model is similarly boosted by 2.1% as the joint-training practice. However, we observe that in most cases, pre-training is slightly inferior to joint training. We conjecture it is due to the high quality of our processed synthetic pairs, making them play a more important role when used by joint training.

## 4.4 Ablation Studies

Unless otherwise specified, we conduct ablation studies on ADE20K with SegFormer-B4.

**Effectiveness of our filtering and re-sampling strategies.** We first demonstrate the indispensable role of our proposed filtering and re-sampling strategies by examining the transferring performance

| Training Scheme | Mask2Former + Swin-T | SegFormer + MiT-B4 | Segmenter + ViT-S |
|---|---|---|---|
| Real Only ($1\times$ iters) | 48.7 | 48.5 | 46.2 |
| Real Only ($2\times$ iters) | 49.2 | 48.6 | 46.7 |
| Real + Synthetic | **52.0 (↑ 2.8%)** | **50.6 (↑ 2.0%)** | **47.9 (↑ 1.2%)** |

Table 8: We attempt to double the total iteration for real-image training paradigm ($2\times$ iters). It is evidently inferior to the "real + synthetic" setup under the same training iterations and batch size.

from the synthetic set, the same as the practice in Section 4.2. As shown in Table 6, on ADE20K, the baseline without processing synthetic images, shows a considerable performance gap with real images (43.3% *vs*. 48.5%). Then with our pixel-level filtering criterion, the result is improved tremendously (43.3% → 47.4%). Our hardness-aware re-sampling principle also remarkably boosts the result from 43.3% to 45.1%. Combining both strategies, the transferring performance achieves 48.3%, outperforming the baseline by 5%. Similar observations also hold true on COCO.

Furthermore, we evaluate our two processing strategies when jointly training synthetic and real images in Table 7. Although the synthetic images look impressive, they can not benefit real images if combined directly without any processing (48.5% → 48.2%). Equipped with our proposed strategies for synthetic images, they successfully boost the fully-supervised baseline from 48.5% to 50.3%.

**Training real images at double budget.** As aforementioned, due to the halved number of real images in each joint training mini-batch, we double the training iterations. Therefore, we similarly double the training iterations for regular real-image training. As shown in Table 8 of ADE20K, doubling the training iterations under the real-image paradigm (real only $2\times$ iters) does not incur noticeable improvement over the baseline (real only $1\times$ iters). More importantly, it is still significantly inferior to our synthetic joint training (real + synthetic) across a wide range of network architectures.

**Performance with respect to the number of synthetic images.** It is reported in [2] that when the number of synthetic images surpasses real images, more synthetic images even cause worse performance. As present in Table 9, if not applying our processing strategies for synthetic images, the observation is similar to [2]. *However*, as long as the two strategies are adopted, the performance will

| $N_{\max}$ | 6 | 10 | 20 |
|---|---|---|---|
| Filtering & Re-sampling ✗ | 43.7 | 43.6 | 43.3 |
| Filtering & Re-sampling ✓ | 47.2 | 47.7 | **48.3** |

Table 9: Ablation studies on the number of synthetic images, which is controlled by $N_{\max}$. The model is trained with synthetic images only.

keep increasing as the synthetic set enlarges. It further proves the necessity of designing appropriate filtering and re-sampling strategies to effectively learn from synthetic images.

**Quantitative comparison with online hard example mining (OHEM).** In Section 3.3, we have discussed the commonality and difference between our hardness-aware re-sampling strategy and OHEM. In Table 10, we demonstrate our method is also empirically superior to OHEM.

| Baseline | + OHEM | + Our Re-sampling |
|---|---|---|
| 44.0 | 44.2 | **45.4** |

Table 10: Comparison between our hardness-aware re-sampling strategy and OHEM with Segmenter-S.

**Hyper-parameter tolerance margin $\alpha$.** As shown in Figure 5, our proposed filtering strategy is robust to the tolerance margin $\alpha$, performing consistently well when $\alpha$ ranges from 1 to 2.

## 5 Conclusion

In this work, we present a new roadmap to enhance fully-supervised semantic segmentation via leveraging densely annotated synthetic images from generative models. We propose two highly effective approaches to better learn from synthetic images, *i.e.*, filtering noisy synthetic regions at the pixel and class level, and sampling more synthetic images for hard-to-learn masks. With our two processing strategies, synthetic images alone can achieve comparable performance with their real counterpart. Moreover, we investigate two training paradigms, *i.e.*, pre-training and joint training, to compensate real images with synthetic images for better fully-supervised performance.

**Acknowledgment.** This work is supported by the National Natural Science Foundation of China (62201484, 62222604), HKU Startup Fund, and HKU Seed Fund for Basic Research. We also want to thank FreestyleNet [71] for providing high-quality generative models.

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

# A  More Implementation Details

Following FreestyleNet [71], we resize all semantic masks to 512x512 for image synthesis. The guidance scale of the diffusion model is set as 2.0, and the sampling step is 50. For better reproducibility, we pre-define and fix a sequence of $N_{\max}$ seeds to synthesize densely annotated images.

For synthetic pre-training, we adopt exactly the same training protocols as real images. Then, during fine-tuning, the base learning rate is decayed to be half of the normal learning rate. Since our whole model parameters are pre-trained with synthetic images, the fine-tuning learning rate is the same throughout the whole model. The model is pre-trained and fine-tuned for the same iterations as real images. For joint training with real and synthetic images, real images are over-sampled to the same scale as synthetic images to make each mini-batch evenly composed of real images and synthetic images. The batch size is the same as real-image training. Each real image is iterated for the same number of epochs as the real-image training paradigm.

As for other hyper-parameters, *e.g.*, data augmentations and evaluation protocols, they are set exactly the same as those in regular training paradigms. We adopt the MMSegmentation codebase for our development. We use $8 \times$ Nvidia Tesla V100 GPUs for our training experiments.

# B  The Most Improved Classes

We list the most improved ten classes on ADE20K (the gain is measured by IoU): (1) ship: +68.19, (2) microwave: +48.72, (3) arcade machine: +45.85, (4) booth: +45.66, (5) oven: +30.86, (6) skyscraper: +23.23, (7) swimming pool: +15.52, (8) armchair: +14.6, (9) hood: +14.43, (10) wardrobe: +13.24.

# C  Discussions for Future Works and Limitations

**Future works.** In this work, we use the off-the-shelf semantic image synthesis model to generate densely annotated images. We have validated that the fully-supervised baseline can be remarkably boosted with these synthetic training pairs. We expect more considerable improvements can be achieved in future works by (1) better-trained or larger-scale pre-trained generative models, (2) larger-scale synthetic training pairs, and (3) taking the class distribution into consideration during image synthesis.

**Limitations.** It is relatively time-consuming to produce synthetic training pairs. For example, it takes around 5.8 seconds to synthesize a single image with a V100 GPU. In practice, we speed up the synthesis process with 24 V100 GPUs. Therefore we can construct the entire synthetic training set for ADE20K and COCO-Stuff in two days. In addition, considering the great potential of our densely annotated synthetic images, it will be more practical to apply our proposed roadmap to real-world scenarios, *e.g.*, medical image analysis and remote sensing interpretation. We plan to conduct these explorations in future works.

# D  Visualization of Densely Annotated Synthetic Images and Filtered Regions

We display our diverse densely annotated synthetic images in Figure 6 of ADE20K and Figure 7 of COCO-Stuff. Besides, we also visualize our filtered regions during training for each synthetic image. It can be observed that there exist several patterns of filtered regions, *e.g.*, boundary regions, synthesis failure cases, and small or rare objects. Please refer to the following pages for details.

Real Image     Semantic Mask        Diverse Synthetic Images & Filtered Region (White Region)

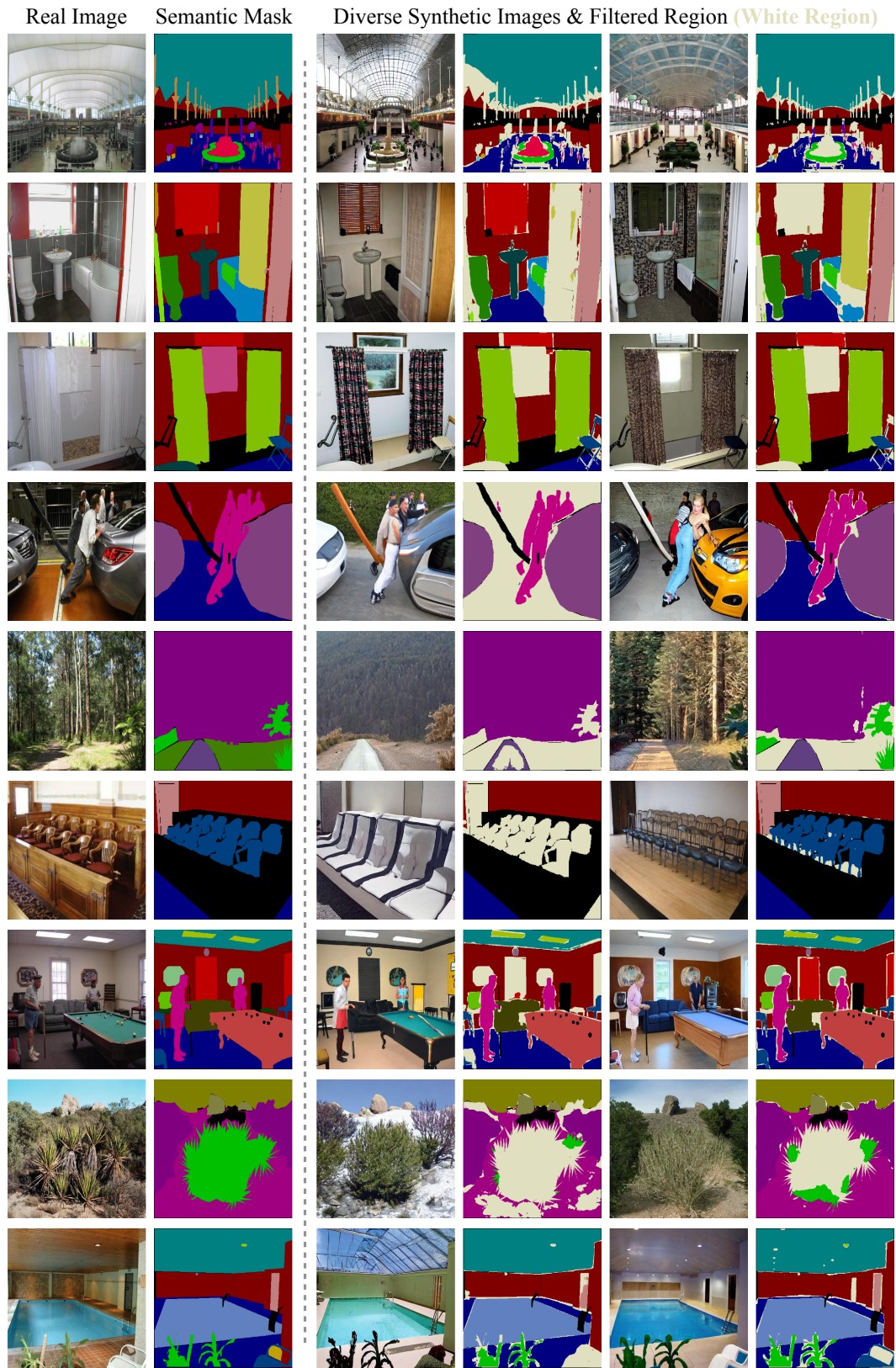

Figure 6: Visualization of diverse densely annotated synthetic images on **ADE20K**, as well as the filtered regions (white regions in the semantic mask). Note that the black regions in the masks are officially marked as "ignored region" by the original ADE20K dataset.

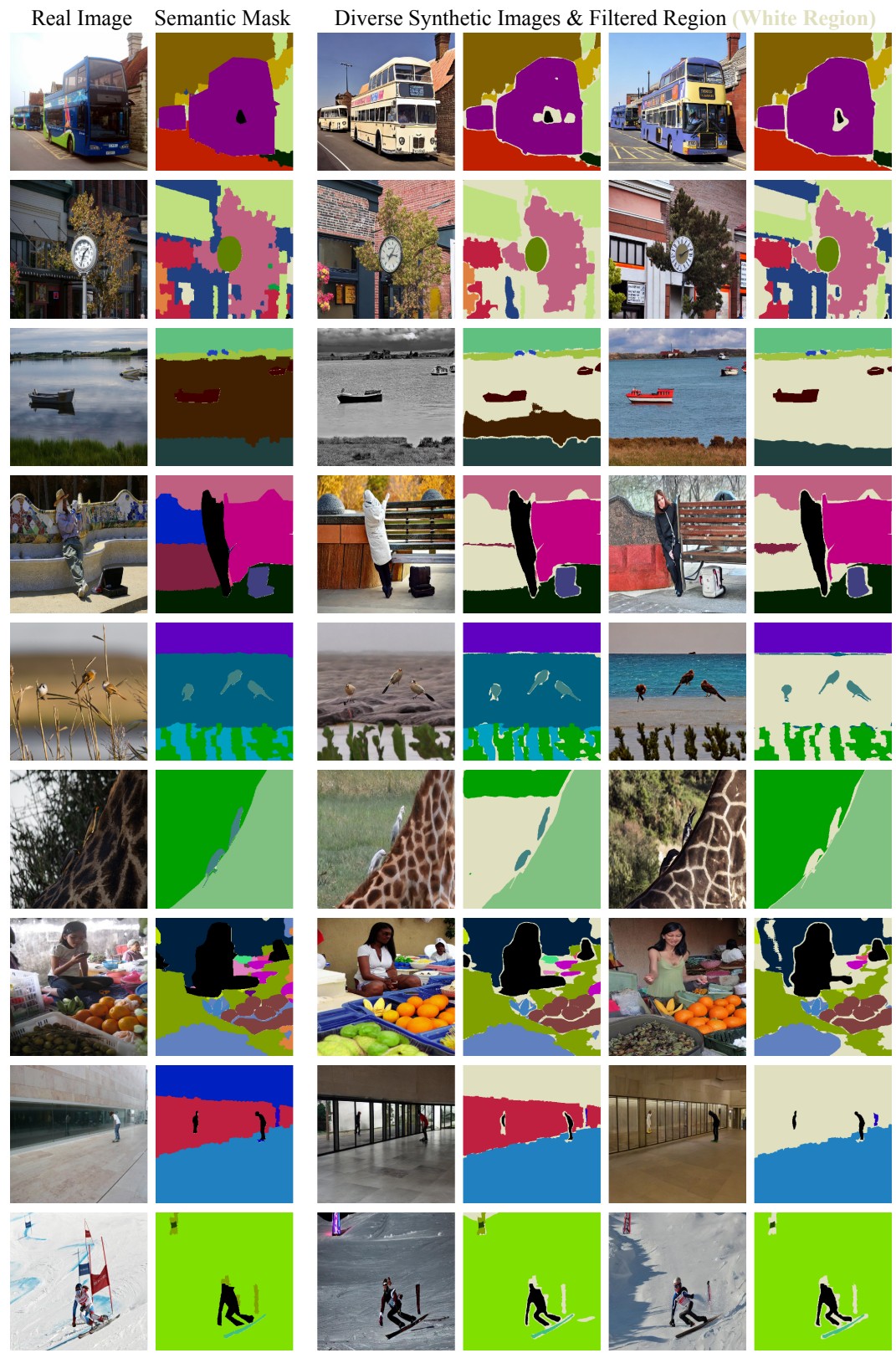

Figure 7: Visualization of diverse densely annotated synthetic images on **COCO-Stuff**, as well as the filtered regions (white regions in the semantic mask).

