# OpenReview forum: "FreeMask: Synthetic Images with Dense Annotations Make Stronger Segmentation Models"
_NeurIPS.cc/2023/Conference — NeurIPS 2023 poster_

### Official Review · Reviewer_VTnr · 2023-06-24

**Soundness:** 3 good
**Presentation:** 3 good
**Contribution:** 2 fair
**Rating:** 4
**Confidence:** 4

**Summary:**

The authors propose to use paired synthetic data to train a semantic segmentation model. Specifically, a filtering strategy and a resampling strategy are proposed to control the quality of synthetic data. In this way, the paired synthetic data could further promote the standard segmentation model.

**Strengths:**

- The overall idea makes sense, and the implementations are reasonable.
- The paper is well written and easy to follow.

**Weaknesses:**

- There may be conflict between filtering hard pixels and re-sampling hard masks, because the re-sampled hard samples may be filtered out. More in-depth analysis could be appended to solve this concern.
- The performance gains  (e.g., 48.5 → 50.6) seems not significant enough, considering that the proposed method introduce too many extra heavy processings. Specifically, 1) fine-tune the generative model on the specific dataset; 2) pre-train a segmentation model, i.e., Line 197; 3) finally train the desired segmentation model.
- The effectiveness of the proposed method depends on the gap between the generative model and specific dataset, and thus the in-depth analysis about the adaption (fine-tuning) of generative model is necessary. On the one hand, the generative model is hard to adapt when the specific dataset is small. On the other hand, the generative model is harder to provide rich cases when the specific dataset is large. Some in-depth analyses could be appended.


**Questions:**

See *Weaknesses.

**Limitations:**

See *Weaknesses.

---

> ### Author Rebuttal · Authors · 2023-08-10
>
> We are sincerely grateful for your efforts and constructive feedback. We hope the concerns are well addressed.
>
> **Q5-1: Conflict between two strategies: the generated additional hard samples are filtered out**
>
> Our filtering strategy does not discard all hard samples. It mainly aims to detect noisy synthetic regions, rather than ignoring all hard samples. As described in L205-207 of our main paper, we set a tolerance margin for filtering, which enables the middle-hardness samples to be kept and mainly removes synthesis failure cases with extremely large losses, as also evidenced by the quantitative results in our response Q2-7 to Reviewer 6xi1. Therefore, the additionally generated images for hard masks are beneficial.
>
> **Q5-2: Improvement (48.5 $\rightarrow$ 50.6) is not significant enough, especially when the method is sophisticated**
>
> **[Our improvement is much larger than model-centric works]** Actually, the +2.1% gain over the fully-supervised performance on ADE20K is already a tremendous improvement. Across our investigated seven architectures (appendix Tab 1), we improve the fully-supervised baseline by +2.0% on average. **In comparison**, the hot research line of model-centric works, *e.g.*, SegViT [NeurIPS'22], only improves its precedent StructToken from 50.9% $\rightarrow$ 51.3% (+0.4%).
>
> **[Our pipeline is straightforward]** Moreover, our framework is straightforward to implement and easy to deploy. We do not have to use synthetic images for pre-training and then fine-tuning. We can simply combine them with real images for joint-training. There is only a single stage of training in this case. We demonstrate the strong results under this joint-training scenario in main Tab 1 and appendix Tab 1. It is also worth noting that, once our synthetic images have been generated for a specific application scenario, they can be saved for later use and benefit various types of model architectures. We will also open-source our large-scale ADE20K and COCO synthetic training images for our community to use.
>
> **Q5-3: How to ensure generation quality with small datasets? Will the effectiveness be diminished on large datasets?**
>
> **In case of small datasets:**
> - **[Intuitive analysis]** Owing to the highly capable pre-trained text-to-image diffusion model, the generation quality is still very promising even with limited fine-tuning images. A good example of this is DreamBooth, which can only use a few (3-5) images to personalize a generation model. More importantly, even if the synthesis quality is not promising in some rare scenes, our proposed filtering strategy can safely ensure the remaining synthetic regions are relatively clean. In fact, considering that small datasets are especially hungry for training data, our delicately processed synthetic set will be precious to them and hopefully enhance the fully-supervised baseline remarkably.
> - **[Quantitative results]** To validate this, we select a subset of 1K images from ADE20K. We fine-tune the Stable Diffusion model only with these limited images, and produce 10K synthetic images (already filtered and re-sampled) with this fine-tuned generator. As a result, merely with 1K real images, the validation mIoU is 22.8%. After jointly trained with our 10K synthetic images, the performance is tremendously boosted from 22.8% $\rightarrow$ 28.6% (**+5.8%**). This improvement indicates our framework can work very well in limited-data scenarios.
>
> **In case of large datasets:** Furthermore, as for the concern about a large real dataset, we have fully demonstrated the effectiveness of our method on the large-scale COCO in main Tab 2, main Fig 5, and appendix Tab 2 (the average improvement is 1.2% across **six** architectures). Note that COCO is one of the largest datasets with challenging taxonomy in semantic segmentation, consisting of 118K training pairs. Lastly, we hope to emphasize that it is extremely difficult and costly to collect million-scale training pairs in semantic segmentation (the latest Segment Anything dataset lacks semantic labels). Therefore, we believe our proposed roadmap of utilizing high-quality synthetic pairs to complement the real dataset is of great value.

---

> > ### Author Response · Authors · 2023-08-14
> > **Looking forward to further feedback**
> >
> > Dear Reviewer VTnr,
> >
> > We are sincerely grateful to you for the precious time and selfless efforts you have devoted to reviewing our paper.
> >
> > We would like to inquire whether our response has addressed your concerns and if you have the time to provide further feedback on our rebuttal. We are more than willing to engage in further discussion.
> >
> > Best regards,
> >
> > Authors of paper 6586.

---

> > ### Author Response · Authors · 2023-08-20
> > **Looking forward to further feedback**
> >
> > Dear Reviewer VTnr,
> >
> > We are sincerely grateful to you for the precious time and selfless efforts you have devoted to reviewing our paper.
> >
> > We have provided our detailed response to each concern. Since the deadline for reviewer-author discussion is approaching, we would like to inquire whether our response has addressed your concerns and if you have the time to provide further feedback on our rebuttal. We are more than willing to engage in further discussion.
> >
> > Best regards,
> >
> > Authors of paper 6586.

---

> > > ### Author Response · Authors · 2023-08-21
> > >
> > > Dear Reviewer VTnr, thank you very much for your precious time and great contributions.
> > >
> > > Since we are approaching the discussion deadline in 10 hours, we would like to ask if there is any further feedback on our rebuttal.
> > >
> > > We are greatly motivated that Reviewer 6xi1 has improved the score from 3 to 5 (borderline accept), Reviewer FZD9 has improved from 4 to 6 (weak accept), and Reviewer jaKB keeps a score of 7 (accept). Besides, we believe our work has improved a lot from your constructive feedback. For example, we will add detailed experiments and discussions about the effectiveness of our method in case of small or large datasets (also provided above in the [rebuttal](https://openreview.net/forum?id=XOotfgPiUF&noteId=d5NhljZDiI)).
> > >
> > > We are eagerly looking forward to your further attention. Thank you very much.
> > >
> > > Best regards,
> > >
> > > Authors of paper 6586.

---

> > > > ### Comment · Area_Chair_jXMB · 2023-08-21
> > > > **Please provide feedback to authors**
> > > >
> > > > Dear Reviewer VTnr,
> > > > what are you concerns after the rebuttal and in light of the other reviews? Could you please provide some feedback to the authors?
> > > > Thanks,

---

### Official Review · Reviewer_jaKB · 2023-07-04

**Soundness:** 3 good
**Presentation:** 4 excellent
**Contribution:** 3 good
**Rating:** 7
**Confidence:** 4

**Summary:**

The paper introduces an automatic dataset generation with mask-to-image translator. The proposed dataset generator enables to generate controllable and consistent semantic labels in generated images. The labels can be treated as fully supervised teachers from a generator and create pre-trained segmentation models under the (synthetic) supervised learning. Moreover, the dataset generation framework serves difficulty levels inside of the contents in an image. In the framework, the authors utilize FreestyleNet pre-trained with StableDiffusion in order to translate from mask to image (mask-to-image) for the pre-training pairs.

**Strengths:**

- This paper is clearly written and easy to understand. The presentation to describe the proposed method (e.g., Figures 2, 3, 4) is high quality and convincing in visualization.

- The paper could serve as a good example for data-driven approach in semantic segmentation tasks.

- Experiments and their results cover multiple aspects of the performance of the synthetic pre-training. The paper will serve a good inspiration to others in synthetic pre-training and related topics.

- The two different aspects, 'noise filtering (Section 3.2)' and 'image re-sampling (Section 3.3)' in synthetic pre-training are very reasonable approaches. Indeed, these two have been shown to be effective as shown in Tables 4 and 5. These two approaches are complementarily improve the segmentation performance in terms of mIoU.

**Weaknesses:**

The reviewer does not find a critical weakness in the paper. However, it could be shown a little weaknesses.

- The paper includes the descriptions as 'controllable' in l.38 and l.179, however, could the proposed approach make more controllable dense annotations? For example, a semantic mask can be manually edited in addition to the image dataset as is. Could the authors consider the kind of flexibly edited approach by humans? As an example of image editing and generation, the reviewer can raise GauGAN from [Park+, CVPR19]. The reviewer doesn't think any image editing should be acceptable, but it would be effective in terms of making the synthetic segmentation dataset more flexible.

[Park+, CVPR19] Taesung Park et al. "Semantic Image Synthesis with Spatially-Adaptive Normalization" in CVPR 2019.

- This is not a critical weakness, however, can alone with the synthetic pre-training surpass the real-image dataset? In fact, the single approach with synthetic images can reach comparative scores as shown in Tables 1 and 2. In some cases, the gap is quite close each other (e.g., 48.5 vs. 48.3 in SegFormer-B4 on ADE20k dataset). If it is possible on paper limitation, adding the discussion would make the paper more valuable.

**Questions:**

- Can the proposed dataset generator with FreestyleNet precisely generate the training images by taking care of object boundaries? As the paper described in l.144-145, "We observe it is more precise in structures to synthesize images from masks than predict masks for images, especially in boundary regions". However, the reviewer is concerning that the quality of object boundaries may affect the recognition accuracy even if the process passed as mentioned in Section 3.2 filtering noisy synthetic regions.

**Limitations:**

There are no negative limitations and societal impacts. On the contrary, the paper alleviate the privacy issues by means of synthetic image (pre-)training. The direction can effectively accelerate the kind of ethical problems in the future.

---

> ### Author Rebuttal · Authors · 2023-08-10
>
> We are sincerely grateful for your appreciation of our work. Many thanks for your efforts and constructive feedback. We hope the concerns are well addressed.
>
> **Q4-1: How can the proposed method be more "controllable"?**
>
> Thank you. We claim our framework of learning from synthetic images is more controllable, because we believe that in future works, we can edit the semantic masks to construct a targeted synthetic set. For example, considering the class imbalance issue, we may use *Copy-Paste on semantic masks* to produce new layouts that are inclined to rare classes to re-balance the class distribution. During the process, we also need to take the structure and object co-concurrence into account. In addition, we, or other works, may design a new generator for mask synthesis, which is conditioned on class distributions directly.
>
> **Q4-2: Can synthetic images surpass real images alone?**
>
> Since current results achieved by synthetic images are extremely close to real images, we expect synthetic images alone can surpass real images in the future when (1) more powerful image generators are utilized and (2) more effective processing strategies are proposed for synthetic images.
>
> **Q4-3: The quality of boundary alignment**
>
> Thank you. As shown in Fig 1 of our main paper and according to our manual check, we find the boundary alignment between synthetic images and conditioned masks is highly precise. The alignment quality is much higher than widely adopted pseudo labeling strategies that predict masks from input images. We safely conjecture that this is because our input semantic masks are encoded in the discrete one-hot format, which is "sharp" and much easier to recognize boundaries than the "smooth" RGB values that DatasetGAN feeds.

---

> > ### Comment · Reviewer_jaKB · 2023-08-21
> > **Response to author rebuttal**
> >
> > Thank you so much for the authors. The Q&A is one of the most important things in the future generative AI and learning. The reviewer encourage the authors, and will keep the paper rating as "7: Accept". Thanks again for the discussion.

---

> > > ### Author Response · Authors · 2023-08-21
> > >
> > > Dear Reviewer jaKB,
> > >
> > > We are deeply grateful for your great appreciation of our work. Thank you so much for all the precious time you have devoted and the constructive feedback you have provided. We will include all your feedback in our final version.
> > >
> > > Best regards,
> > >
> > > Authors of paper 6586.

---

### Official Review · Reviewer_FZD9 · 2023-07-05

**Soundness:** 3 good
**Presentation:** 3 good
**Contribution:** 2 fair
**Rating:** 6
**Confidence:** 5

**Summary:**

The Paper talks about using synthetic images generated using generative models as the training set to achieve stronger semantic segmentation models. The efficacy of the model is evaluated on ADE2K and COCO dataset, using SegFormer model. Authors propose pretraining with synthetic and joint images to evaluate which one works the best and in which scenario. Uses sampling and filtering training mechanism to further help the results.

**Strengths:**

1. The paper demonstrates a clear structure, effectively explaining the fundamental modules and training procedure involved. It successfully addresses the task of scene understanding by leveraging annotations obtained from generative models, thereby enhancing the results of the semantic segmentation task.

2. The inclusion of a self-adaptive module within the training pipeline is a valuable contribution. This module effectively refines erroneous or spurious training examples, leading to improved model performance.

3. The paper incorporates a sampling strategy that focuses on hard-to-learn cases. By prioritizing these challenging instances, the overall performance of the model is enhanced, resulting in better final results.

**Weaknesses:**

Methods used to improve the performance.
1. Mining Hard Examples: This is not something novel and (different flavors of it) has been used in many previous work to help in training better examples in case semantic segmentation and object detection.

2. Remove extraneous and harmful samples: Again the filtering mechanism proposed here is not that novel as previous works in the domain of active learning have again used similar techniques to improve training.

3. Use off-the-self models both for generation and training the model, nothing novel in that regard.

**Questions:**

1. Would be interesting to know, what is the class frequency for the new generated  dataset used to train the model. Does it follow the similar class frequency distribution as seen in ADE20K and COCO?

2. What are the classes for which the improvement is seen to be significant?

---

> ### Author Rebuttal · Authors · 2023-08-10
>
> We are sincerely grateful for your efforts and constructive feedback. We hope the concerns are well addressed.
>
> **Q3-1: Mining hard examples are not novel**
>
> We are only related to existing works (*e.g.*, OHEM) *from the aspect of motivation*. It is indeed a widely shared motivation. However, we use totally different approaches. We produce more synthetic images for harder masks, which is fundamentally different from OHEM that simply ignores high-confidence pixels. We allocate synthesis quotas discriminatively, while OHEM only performs selection. Our designed method is well suited to our framework that aims to learn from synthetic data. You may also refer to our response Q2-1 to Reviewer 6xi1 for more details. In Q2-1, we also quantitatively demonstrate the superiority of our re-sampling strategy over OHEM.
>
> **Q3-2: Compare our filtering strategy with the arts in active learning**
>
> Our filtering strategy *discards noisy synthetic regions* (failure cases during synthesis), while active learning aims to find the most informative samples for humans to annotate. The most informative samples in active learning mostly exhibit large or middle losses, while large-loss samples in our scenario are noisy and need to be discarded. Therefore, our motivations are totally different, even contrary to active learning.
>
> **Q3-3: Using off-the-shelf models both for generation and training the model is not novel**
>
> We hope to highlight that we are indeed a pioneering work to improve the *fully-supervised semantic segmentation performance* with *synthetic data*. Our motivation and our framework are novel in several aspects. We have thoroughly compared our work with previous works in Related Work--"Learning from synthetic images". **[Differences]** Briefly summarized here:
> - **[Few-shot *vs*. fully-supervised]** Existing works only focus on a constrained scenario, *e.g.*, few-shot labels, while we address the challenging but widely acknowledged fully-supervised scenario.
> - **[Classification *vs*. semantic segmentation]** Existing works mostly address the classification task, which is cheap to label and even not so urgent for human labels (unsupervised learning performs quite well), while our targeted semantic segmentation task is highly expensive and laborious to annotate.
> - **[Image-to-pseudo-mask *vs*. mask-to-image]** Existing works predict pseudo masks for synthetic images, which is not precise enough, while we adopt a mask-to-image synthesis pipeline to produce better aligned image-mask pairs, as evidenced by our response Q2-5 to Reviewer 6xi1.
> - **[Blindly using *vs*. carefully processing synthetic data]** Existing works of learning from synthetic data ignore the importance of processing synthetic data discriminatively, while our proposed filtering and re-sampling can impressively improve the effectiveness of synthetic data and ultimately yield a much stronger model.
>
> We provide more discussions in the global response. Please refer to it if you are still concerned. Thank you very much.
>
> **Q3-4: Class frequency of synthetic set and real set**
>
> Since we synthesize images conditioned on semantic masks from the real training set, the class frequency of our synthetic set is exactly the same as the real set if no filtering and re-sampling strategies are applied. Then, if we apply filtering and re-sampling strategies to the synthetic set, its class frequency will be changed. Please refer to our uploaded global PDF for detailed visualizations of the class frequency. Thank you.
>
> **Q3-5: The most improved classes**
>
> We list the most improved ten classes on ADE20K (the gain is measured by IoU): (1) ship: +68.19, (2) microwave: +48.72, (3) arcade machine: +45.85, (4) booth: +45.66, (5) oven: +30.86, (6) skyscraper: +23.23, (7) swimming pool: +15.52, (8) armchair: +14.6, (9) hood: +14.43, (10) wardrobe: +13.24.

---

> > ### Author Response · Authors · 2023-08-14
> > **Looking forward to further feedback**
> >
> > Dear Reviewer FZD9,
> >
> > We are sincerely grateful to you for the precious time and selfless efforts you have devoted to reviewing our paper.
> >
> > We would like to inquire whether our response has addressed your concerns and if you have the time to provide further feedback on our rebuttal. We are more than willing to engage in further discussion.
> >
> > Best regards,
> >
> > Authors of paper 6586.

---

> > > ### Comment · Reviewer_FZD9 · 2023-08-18
> > > **Reply to the Authors**
> > >
> > > Thank the authors for providing response in detail. I would like to agree with some of the answers that authors have responded with. About the improvement seen for the classes. The observation seems like it improves the results for some of the more less seen classes than the more often seen one at least for ADE20K dataset. My follow up questions would be
> > >
> > > 1. What could be probable reasons for improvement seen in the classes such as  ship: +68.19, (2) microwave: +48.72, (3) arcade machine: +45.85, (4) booth: +45.66, (5) oven: +30.86, (6) skyscraper: +23.23, (7) swimming pool: +15.52. There does not seem to be any relation in terms of size or semantics. What are authors comment on this ?
> > >
> > > 2. Also for both COCO and ADE20K datasets, it would be interesting to see what is the improvement seen for classes which seen the most.

---

> > > > ### Author Response · Authors · 2023-08-18
> > > > **Further response**
> > > >
> > > > Dear Reviewer FZD9, thank you very much for your further feedback.
> > > >
> > > > For the first question about the large improvements of our listed classes, we think there are three reasons jointly contributing to this:
> > > >
> > > > - **[More diverse data distributions]** Our processed synthetic training pairs largely compensates the original real training dataset. For example, there are 11x more pixels belonging to *swimming pool* after introducing our synthetic set. These additional novel training samples sufficiently provide more diverse data distributions for our new training set, enabling stronger test performance and even stronger out-of-distribution (OOD) ability (as for OOD, we politely refer you to our response Q2-9 to Reviewer 6xi1).
> > > > - **[Larger room to improve]** To be honest, there is much larger room to improve on these classes. For example, the baseline IoU (trained merely with real images) is only 10.82% on the *ship* class, 36.19%  on the *arcade machine* class, and 30.06% on the *booth* classes.
> > > > - **[Promising synthesis quality]** These classes are indeed not rare in the training set of our pre-trained Stable Diffusion (SD) model. Thus, the synthesis quality of these classes is acceptable. For example, according to our search, there are 15,473 *ship*-relevant images and 5,431 *swimming-pool*-relevant images on the SD training set LAION-5B. After further fine-tuned on our target set (*e.g.*, ADE20K), the synthesis ability can be further enhanced.
> > > >
> > > > For the second question, the most frequent class on ADE20K is *wall*, and we improve its IoU from 77.51% to 78.65%. The most frequent class on COCO is *person*, and we improve its IoU from 85.15% to 85.95%. These two improvements are already significant, because the real training samples are already abundant on the two classes and the baseline results are rather competitive (classes with nearly 80% IoU are very challenging to improve under the same model architecture, like the Pascal VOC and Cityscapes benchmarks).
> > > >
> > > > We sincerely appreciate your precious time and further feedback. We are wondering whether our further responses have fully addressed your concerns. Thank you very much.

---

> > > > > ### Comment · Reviewer_FZD9 · 2023-08-18
> > > > > **Reply to Authors**
> > > > >
> > > > > Thank you for providing further details about the quantitative results on class specific results. It is good to know that "there are 11x more pixels belonging to swimming pool after introducing our synthetic set. " It would be good to add these statistics for most improved classes for both ADE2K and COCO dataset in the final version. Giving us a sense about how exactly does this impact the final performance. Given the final response I would change my rating to Borderline Accept.

---

> > > > > > ### Author Response · Authors · 2023-08-18
> > > > > > **Thank you for your appreciation of our response**
> > > > > >
> > > > > > Dear Reviewer FZD9, thank you for acknowledging our response and appreciating our work. We will follow your advice and draw a detailed bar chart to demonstrate the class-wise increased sample number and class-wise IoU improvement in our final version. Thank you for being willing to change your rating to Borderline Accept.
> > > > > >
> > > > > > But it seems the score has not been modified. Would there be any further concerns we can help?

---

> > > > > > > ### Comment · Area_Chair_jXMB · 2023-08-21
> > > > > > >
> > > > > > > Dear authors,
> > > > > > > with FZD9's raise to weak accept I consider the concerns addressed and your question answered.

---

### Official Review · Reviewer_6xi1 · 2023-07-07

**Soundness:** 3 good
**Presentation:** 3 good
**Contribution:** 2 fair
**Rating:** 5
**Confidence:** 5

**Summary:**

This paper proposes to generate densely annotated synthetic images with generative models to help supervise the learning of fully supervised semantic segmentation frameworks. To improve the effectiveness of synthetic images, the authors further design a robust filtering criterion to suppress noisy synthetic samples at the pixel and class levels and propose an effective metric to indicate the hardness of semantic masks where they sample more synthetic images for harder masks. Ablation studies validate the effectiveness of the proposed method.

**Strengths:**

1. The logic of the article is generally clear, and the method is easy to understand.
2. The ablation experiments of this paper indicate the effectiveness of the proposed method to a certain extent.

**Weaknesses:**

1. The proposed strategy termed re-sampling synthetic images based on mask-level hardness is somehow like “Online Hard Example Mining” (OHEM) which is widely used in computer vision area including semantic segmentation, and the adopted metric to measure the sample hardness is also the widely used average losses of all pixels in the input image. Where is the novelty of this part? I don't think generating more hard samples should be the contribution of this section since it is more likely to belong to the contribution of Section 3.1.
2. The analysis in Section 4.4 for Table 6 is not convincing in my eyes. First, after adopting “Filtering & Re-sampling”, N_max no longer denotes for the number of synthetic images used for training, thus the comparison in Table 6 is unreasonable. If I understand correctly, the number of used synthetic images should be n_p in Eq. (2), which may result in a wrong conclusion in Line 339-348. Second, where is the performance ceiling when setting N_max after adopting “Filtering & Re-sampling”? It looks like setting larger N_max (> 20) will yield better segmentation results. Finally, how to set p in Eq. (2). The reviewer does not find this detail in both paper and the supplement materials.
3. Why there are no quantitative comparisons between the proposed method and previous methods like DatasetGAN and BigDatasetGAN? The authors argue that “the main drawback of such methods is the involvement of expensive human efforts.” So, how about applying a simple pseudo labeling strategy on these generated images and comparing the results between image-pseudo labeling strategy and your proposed mask-to-image synthesis strategy? If the results are comparable, where are the advantages of using your proposed method? The reviewer believe it is important to compare your method and previous similar methods quantitatively to show the novelty of this paper.
4. It seems that the performance improvements in Table 3 for Mask2Former is limited. To my knowledge, just simply run Mask2Former for two times may also bring such improvements. Could the author give some analysis about the limited gains for Mask2Former?
5. The filtering strategy in Section3.2 is naïve and not reliable in my eyes. Specifically, could the authors provide any quantitative results to show that the proposed filter strategy could filter noisy synthetic region rather than some hard samples?
6. Previous studies like “Focal Loss for Dense Object Detection” indicate that OHEM is unreasonable. Whether the authors compare your re-sampling strategy with some objective-function-based strategy like Focal Loss, weighted cross entropy loss to show the effectiveness of your method?
7. Why higher mIoU must be a good point is Table 4? For example, if your method could generate some densely annotated synthetic images with other domains which is different from the training and testing image? Whether it would lead to the decrease of mIoU but make the segmentor adapt to broader application scenarios? The reviewer thinks the latter is more important.
8. Could the authors give the results of performing the methods in Section 3.2 and 3.3 on the real training images? I mean filter the real images with the proposed strategies and re-train the model to see the importance of the generated images.

**Questions:**

Questions are listed in the weaknesses.

**Limitations:**

Limitations have been discussed in the paper.

---

> ### Author Rebuttal · Authors · 2023-08-10
>
> We are sincerely grateful for your efforts and constructive feedback. We hope the concerns are well addressed.
>
> **Q2-1: Our re-sampling strategy is similar to OHEM**
>
> In L236-L241, we have compared our re-sampling method with OHEM. In semantic segmentation, OHEM ignores high-confidence pixels and only computes the average loss on low-confidence pixels.
>
> **[Relation]** Our motivations are related, *i.e.*, both emphasizing hard samples (however, our hard samples are semantic masks, while OHEM is image pixels).
>
> **[Difference]** Our practices are fundamentally distinguished. We "generate" additional hard samples for models to sufficiently learn, while OHEM essentially only performs a one-hot "re-weighting" (loss weight 0 for high-confidence pixels, and weight 1 for low-confidence pixels).
>
> **[Superiority]** We *quantitatively* compare our re-sampling method with OHEM below, proving our method is evidently superior to OHEM. (Results below are obtained by training solely on synthetic images with Segmenter-ViT-S, and the filtering strategy is applied to all methods).
>
> |Baseline|+ OHEM (thresh=0.7, min kept=100K)|+ Our re-sampling|
> |:-:|:-:|:-:|
> |44.0|44.2|**45.4**|
>
> **Q2-2: How to set $p$ in Eq 2**
>
> As mentioned in L233 "for the $p$-largest-hardness mask", the $p$ denotes *the rank of a mask* in terms of hardness, ranging from 0 to $(N-1)$ as an integer. According to this $p$ (hardness rank), we then determine the synthesis quota (number of synthetic images) for a mask by Eq 2.
>
> **Q2-3: Tab 6 is not convincing, because $N_{\max}$ does not denote the number of synthetic images when using re-sampling**
>
> In the re-sampling case, we actually use fewer synthetic images than the non-re-sampling counterpart, so the better performance from our re-sampling method is convincing. Specifically,
> - when **not using** re-sampling, all semantic masks share the same synthesis quota, *i.e.*, always $N_{\max}$ synthetic images *from a single mask*.
> - when **using** re-sampling, most semantic masks are equipped with fewer than $N_{\max}$ synthetic images. The concrete synthesis quota for each mask is determined by Eq 2, which evenly distributes the number of synthetic images for all semantic masks from 1 to $N_{\max}$. The number of total synthetic images is nearly halved after the re-sampling process. Hence, our superiority over the non-re-sampling counterpart is convincing.
>
> **Q2-4: The performance ceiling when increasing $N_{\max}$**
>
> Please refer to our response Q1-5 to Reviewer 3GaN.
>
> **Q2-5: Comparison with DatasetGAN, ***i.e.***, annotating pseudo semantic masks for synthetic images**
>
> Thank you for your constructive feedback.
>
> (1) Following your advice, we predict pseudo masks for our synthetic images with the SOTA model (ViT-Adapter-BEiTv2-L-IN22K, ICLR'2023) on ADE20K. The mIoU between predicted pseudo masks and GT masks is 49.79. The final validation mIoU is compared below.
> |Image-to-Pseudo-Mask|Mask-to-Image (Ours)|
> |:-:|:-:|
> |44.0|**45.4**|
>
> (2) Besides, as an extension, we further borrow the real COCO data as an unlabeled source for ADE20K. We validate whether COCO images along with the pseudo labeling strategy can benefit our targeted ADE20K. The results are listed below (\*: reproduced by us).
> |Real Only\* (Segmenter-ViT-S)|Real + COCO (Pseudo labeling)|Real + Synthetic (Ours)|
> |:-:|:-:|:-:|
> |45.8|46.2|**48.0**|
>
> **Q2-6: Improvement with Mask2Former-Swin-L-22K is not significant (56.0 $\rightarrow$ 56.4)**
>
> As noted in Tab 3 caption, this specific model is sufficiently pre-trained on the extremely large-scale ImageNet-22K. Therefore, its hunger for downstream fine-tuning data is diminished. For other Mask2Former models, our framework performs impressively, *e.g.*, **+3.3%** with Swin-T and **+1.6%** with Swin-S (appendix Tab 1 \& 3).
>
> **Q2-7: Could the filtering strategy filter noisy synthetic regions rather than hard samples?**
>
> We present average losses on filtered/non-filtered real/synthetic regions below. First, we compare the loss on non-filtered regions (id 1 and 2), the loss magnitude of real and synthetic regions are close. Then when we compare the loss on filtered regions (id 3 and 4), it is obvious that the synthetic loss becomes much larger (nearly 3$\times$) than the real loss. Thus, we can conclude there must exist abundant noise in the filtered synthetic regions.
> |Real non-filtered (id: 1) |Syn non-filtered (id: 2)|Real  filtered (id: 3)|Syn filtered (id: 4)|
> |:-:|:-:|:-:|:-:|
> |0.281|0.363|1.100|3.317|
>
> **Q2-8: Compare our re-sampling with objective functions, ***e.g.***, Focal loss and weighted CE loss**
>
> Results are listed below, where our re-sampling strategy performs much better than the mentioned objective functions.
>
> |Baseline|+ Focal loss|+ Weighted (by class frequency) CE |+ Weighted (by val IoU) CE|Our re-sampling|
> |:-:|:-:|:-:|:-:|:-:|
> |44.0|43.9|37.8|43.9|**45.4**|
>
> **Q2-9: Why higher mIoU in Tab 4 is better, what if considering the robustness?**
>
> We manage to examine the capability of our model to deal with unseen domains. Concretely, we transfer our model trained on ADE20K to COCO. We measure mIoU on the 46 overlapped classes. As proved below, our two strategies are still effective for images from unseen domains.
> |Baseline|+ Filtering |+ Re-sampling|+ Filtering \& Re-sampling|
> |:-:|:-:|:-:|:-:|
> |37.7|40.3|39.4|**41.3**|
>
> **Q2-10: Apply the two strategies to real images**
>
> As for our re-sampling strategy, since there is only a single real image available for a semantic mask, we resort to over-sample (repeat) more real images for harder masks, whose number is decided by Eq 2. As shown below, our re-sampling strategy can still boost the real dataset. And it is expected that the filtering practice downgrades the result on real dataset, because the real dataset is free from noise. This further demonstrates that our filtering design is well suited to synthetic data.
> |Baseline|+ Re-sampling|+ Filtering |+ Filtering \& Re-sampling|
> |:-:|:-:|:-:|:-:|
> |45.8|**46.7**|43.1|43.5|

---

> > ### Author Response · Authors · 2023-08-14
> > **Looking forward to further feedback**
> >
> > Dear Reviewer 6xi1,
> >
> > We are sincerely grateful to you for the precious time and selfless efforts you have devoted to reviewing our paper.
> >
> > We would like to inquire whether our response has addressed your concerns and if you have the time to provide further feedback on our rebuttal. We are more than willing to engage in further discussion.
> >
> > Best regards,
> >
> > Authors of paper 6586.

---

> > ### Comment · Reviewer_6xi1 · 2023-08-17
> >
> > Thanks for authors' answers. I'd like to improve my score as Borderline Accept.

---

> > > ### Author Response · Authors · 2023-08-17
> > > **Thank you for your appreciation of our work**
> > >
> > > Dear Reviewer 6xi1,
> > >
> > > Thank you very much for your acknowledgment of our rebuttal and appreciation of our work. Our work has improved a lot from your constructive feedback. Thank you.
> > >
> > > Best regards,
> > >
> > > Authors of paper 6586.

---

### Official Review · Reviewer_3GaN · 2023-07-10

**Soundness:** 3 good
**Presentation:** 3 good
**Contribution:** 2 fair
**Rating:** 4
**Confidence:** 4

**Summary:**

This paper proposes a method of synthesizing training images and corresponding semantic masks for training a semantic segmentation network. The off-the-shelf semantic image synthesis model, FreestyleNet, is used to generate images from existing semantic masks. Following the proposed re-sampling technique based on mask-level hardness, harder samples are more frequently generated. During training, to avoid the noisy pixel hampering the model training, pixel-level ignoring technique is used. The generated synthetic images are shown to be effective for model training when they are used together with the existing fully supervised labels.

**Strengths:**

- The paper is overall well-written and easy to understand.

- It is interesting that the synthesized images can improve the performance together with fully supervised dataset. This can be practically utilized for many researchers.

**Weaknesses:**

- The proposed method heavily depends on trained mask-to-image generative models. The authors showed that naive generation of synthetic images is not sufficient for training a segmentation network, but the filtering and re-sampling techniques are quite naive. Specifically, ignoring uncertain pixels during training is popularly used for label-efficient learning (e.g., weakly and semi-supervised semanic segmentation).

- Closely related references, copy-paste methods (e.g., [ref1]), are missing. They also augment training data by synthesizing images, but unlike the proposed method, they do not require any additional heavy models. The copy-paste method should be discussed and compared.

[ref1] Simple Copy-Paste is a Strong Data Augmentation Method for Instance Segmentation

- In Abstract, the authors mentioned that "We surprisingly observe that, merely with synthetic images, we already achieve comparable performance with real ones", but I think it is overstated. To synthesize these images, the trained FreestyleNet is required, but FreestyleNet is already trained with real image-mask pairs. In addition, all the technical design (filtering, re-sampling) and values of hyper-parameters are determined with fully supervised validation data. I recommend the authors to tone down the sentence in Abstract.

- I guess the global hardness in Line 229 do not consider the difficulty of segmenting small objects. Intuitively thinking, small objects of hard class should increase the global hardness, but they actually slightly contribute to the global hardness.

- The authors used only the limited number of synthesized images due to the disk issue. I recommend the two additional experiments: 1) the performance change by varying the number of synthesized images. With this trend, we can infer if more synthesized images can further improve the performance. 2) Increase the total number of synthesized images by saving low-resolution images or high-tolerence polygon of masks.


**Questions:**

- Do the authors have a plan to release the code? I strongly recommend the authors to make their code publicly available.



**Limitations:**

No limitation is discussed.

---

> ### Author Rebuttal · Authors · 2023-08-10
>
> We are sincerely grateful for your efforts and constructive feedback. We hope the concerns are well addressed.
>
> **Q1-1: Novelty of our filtering and re-sampling strategies.**
>
> Please refer to our global response for clarification on the filtering strategy.
>
> Please refer to our response Q2-1 to Reviewer 6xi1 for clarification on the re-sampling strategy. Thank you.
>
> **Q1-2: Discussion with Copy-Paste methods**
>
> Thank you for your constructive advice. We agree with you that we should include mixing-based augmentation methods, *e.g.*, Copy-Paste and CutMix, in our related works. We will provide thorough discussions and comparisons in the revised version.
>
> **[Differences]** Existing mixing-based methods fail to generate realistic images, due to ignoring semantic layouts and co-occurrence. Besides, their *re-assembled* images do not contain any novel objects. In contrast, as supported by our visualizations, our method synthesizes extremely realistic novel images and objects, benefiting the perception task significantly.
>
> **Q1-3: Overstatement about "merely using synthetic images is comparable with real images"**
>
> Thank you for your kind reminder. We will weaken the tone in the revised version.
>
> **Q1-4: Small but hard objects contribute less to mask-level hardness**
>
> Thank you for pointing it out. It is true that small but hard objects currently contribute less to mask-level hardness. We conjecture it will be beneficial if these objects can be correctly handled. However, our work aims to propose a *universal* framework to assist real images *across all classes and all objects*. We hope to highlight the value of synthetic data to high-level perception task, especially in the diffusion-model era. We demonstrate the value of simple data filtering and re-sampling principles to synthetic data, *even without taking special cases into account*. In our future works, we will follow your advice to tackle small objects discriminately.
>
> **Q1-5: Performance change with respect to more synthetic images**
>
> After submission, we have attempted to increase $N_{\max}$ from 20 to 40, which means each semantic mask corresponds to at most 40 synthetic images. However honestly, we do not observe an evident improvement. The results under $N_{\max}=20$ and $N_{\max}=40$ are similar. We think there are at least three reasons for this phenomenon:
> - **[Extremely challenging scenario]** We aim to boost the fully-supervised baseline, which is significantly more challenging than the few-shot baseline as existing works do, *e.g.*, DiffuMask. We believe it is more practical in the real world to improve such a challenging but widely acknowledged baseline.
> - **[Remarkable improvements have been achieved by $N_{\max}=20$]** As shown in appendix Tab 1, we remarkably improve the fully-supervised baseline from 48.7% $\rightarrow$ 52.0% (**+3.3%**) on ADE20K with Mask2Former-Swin-T. As a comparison, the research line of model-centric works, *e.g.*, SegViT [NeurIPS'22], only improves its precedent StructToken from 50.9% $\rightarrow$ 51.3% (+0.4%). Across our investigated **seven** architectures (appendix Tab 1), we improve the fully-supervised baseline by **+2.0%** on average, which is a much larger gain than existing model-centric works achieve (mostly only 1%, requiring many trials and errors on model designs).
> - **[Small-scale semantic masks]** We synthesize images conditioned on limited semantic masks from the real dataset (20K masks on ADE20K). These masks are small-scale and not diverse enough. Thus, multiple synthetic images from a shared mask may be redundant and not informative enough. This makes extra synthetic images bring limited further gain. However, as a pioneering work to enhance the challenging fully-supervised semantic segmentation with synthetic data, we think it is acceptable that there is still some room for subsequent works to refine these designs. In the future, there may be some ways to first generate novel and diverse semantic masks for later image synthesis.
>
> **Q1-6: Code release**
>
> We promise to release all our codes, well-trained models, and training logs upon acceptance.
>
> **Q1-7: No limitation is discussed**
>
> We have indeed discussed it in Appendix Section D. We will prioritize it to the main paper in the revised version.

---

> > ### Author Response · Authors · 2023-08-14
> > **Looking forward to further feedback**
> >
> > Dear Reviewer 3GaN,
> >
> > We are sincerely grateful to you for the precious time and selfless efforts you have devoted to reviewing our paper.
> >
> > We would like to inquire whether our response has addressed your concerns and if you have the time to provide further feedback on our rebuttal. We are more than willing to engage in further discussion.
> >
> > Best regards,
> >
> > Authors of paper 6586.

---

> > > ### Author Response · Authors · 2023-08-18
> > > **Further feedback**
> > >
> > > Dear Reviewer 3GaN,
> > >
> > > Thank you for your selfless efforts. As for your previous concern about discussions with Copy-Paste (our response Q1-2), in addition to our previous analysis and qualitative comparisons, we here provide more quantitative results about Copy-Paste (object-level mixing, we use ClassMix in semantic segmentation for no bounding box information) and CutMix (random rectangle-region mixing):
> > >
> > > | Real Only | Real + Copy-Paste | Real + CutMix | Real + Synthetic (Ours) |
> > > |:---------:|:-----------------:|:-------------:|:-----------------------:|
> > > |   45.8    |        45.6 (-0.2)       |     45.9 (+0.1)      |         **48.0 (+2.2)**        |
> > >
> > > In conclusion, Copy-Paste and CutMix do not help much in the *fully-supervised semantic segmentation* task. Similar observations are also reported in [1, 2] (please refer to their provided *fully-supervised* results, the *fully-supervised* results are even downgraded after applying CutMix or ClassMix).
> > >
> > > [1] Semi-supervised semantic segmentation needs strong, varied perturbations, In *BMVC*, 2020.
> > >
> > > [2] ClassMix: Segmentation-based data augmentation for semi-supervised learning, In *WACV*, 2021.

---

> > > > ### Author Response · Authors · 2023-08-20
> > > > **Looking forward to further feedback**
> > > >
> > > > Dear Reviewer 3GaN,
> > > >
> > > > We are sincerely grateful to you for the precious time and selfless efforts you have devoted to reviewing our paper.
> > > >
> > > > We have provided our detailed response to each concern. Since the deadline for reviewer-author discussion is approaching, we would like to inquire whether our response has addressed your concerns and if you have the time to provide further feedback on our rebuttal. We are more than willing to engage in further discussion.
> > > >
> > > > Best regards,
> > > >
> > > > Authors of paper 6586.

---

> > ### Comment · Reviewer_3GaN · 2023-08-20
> >
> > I appreciate the authors' response. My minor concerns are addressed, but I still think the strength of the proposed method comes largely from the superiority of FreestyleNet. The filtering and hard negative sampling seems trivial to me. I would keep my original rating

---

> > > ### Author Response · Authors · 2023-08-20
> > >
> > > Dear Reviewer 3GaN, thank you very much for your further feedback. It is glad to know part of the concerns have been addressed. As for the remaining concern about whether the strength of our method comes largely from FreestyleNet instead of our proposed two processing techniques, we would like to provide some further clarifications here.
> > >
> > > - First, directly using the synthetic pairs from FreestyleNet *without any processing* **will not bring any improvement** to the challenging fully-supervised real-image baseline. This has been supported by Table 5 in our main paper. We borrow the results below (the real-only result is borrowed from main Table 3 with SegFormer-B4). The inferior result of blindly integrating synthetic images (FreestyleNet, -0.2 mIoU) also explains why existing works [1, 2] mostly only address the few-shot scenario, which contains a few real images and is easy to boost. In comparison, with our filtering and hardness-aware synthesis strategies, we can significantly boost the challenging but practical fully-supervised scenario (**+1.8 mIoU**).
> > >
> > > |    | Real data | Synthetic data *w/o processing* | Synthetic data *w/ processing* | validation mIoU |
> > > :-: | :-: | :-:| :-:| :-:
> > > |   Real only    |    ✔    |    |         | 48.5 |
> > > |   FreestyleNet  |    ✔   |  ✔  |          | 48.2 (-0.3) |
> > > |   **Ours**  |    ✔   |    |      ✔    | **50.3 (+1.8)** |
> > >
> > > - Second, we have further validated the necessity and superiority of our two strategies in Table 6 of our main paper. Results are borrowed below. We observe that, *without our proposed techniques*, scaling up the number of synthetic images *does not* yield consistent improvement, even gradually downgrading the performance. In addition, our proposed method tremendously outperforms the plain FreestyleNet which does not include any processing techniques. The performance gap can be as large as **+5.0 mIoU**.
> > >
> > > |  $N_{\max}$  (scaling ratio)  | 6 | 10 | 20 |
> > > :-: | :-: | :-:| :-:
> > > |   w/o processing (FreestyleNet)   |   43.7 | 43.6 | 43.3 |
> > > |   **w/ processing (Ours)**  |  47.2 | 47.7 | **48.3** |
> > > | Superiority over FreestyleNet | +3.5 |  +4.1 | **+5.0** |
> > >
> > > - Lastly, as we provided in the [rebuttal](https://openreview.net/forum?id=XOotfgPiUF&noteId=k6Zw1ZF8r9), our hardness-aware synthesis strategy is not trivial. It is much better than online hard example mining (OHEM). We also borrow the comparisons below (44.2 *vs.* 45.4). As for your opinion that similar filtering strategies also exist in label-efficient learning, we agree with this. Indeed, this is a very common motivation. Similar practices also exist in noisy label learning. However, as we clarified in the global response, *we use similar motivations for totally different purposes*. We aim to recognize the synthesis failure cases, which is proved fundamental to the success of later perception task. The necessity of "cleaning" synthetic images is almost completely ignored in previous works [3] that utilize synthetic images to benefit perception tasks. *We believe that, drawing attention to "more effectively" learning from synthetic images, rather than always focusing on better synthesis quality, is also our contribution to the community.*
> > >
> > > |Baseline|+ OHEM (thresh=0.7, min kept=100K)|+ Our hardness-aware synthesis
> > > |:-:|:-:|:-:|
> > > |44.0|44.2|**45.4**|
> > >
> > > Please tell us if you have any further concerns about this response. We are more than willing to provide any further explanations. Thank you very much for your precious time and great contributions.
> > >
> > > [1] He, Ruifei, et al. "Is synthetic data from generative models ready for image recognition?." *ICLR* 2023.
> > >
> > > [2] Wu, Weijia, et al. "Diffumask: Synthesizing images with pixel-level annotations for semantic segmentation using diffusion models." *ICCV* 2023.
> > >
> > > [3] Azizi, Shekoofeh, et al. "Synthetic data from diffusion models improves imagenet classification." *ICCV* 2023.

---

> > > > ### Author Response · Authors · 2023-08-21
> > > >
> > > > Dear Reviewer 3GaN, thank you very much for your previous feedback.
> > > >
> > > > Since we are approaching the discussion deadline in 10 hours, we would like to ask whether there are any further concerns about our work.
> > > >
> > > > There is only one remaining concern about whether the strength of our method comes largely from FreestyleNet. We have provided comprehensive experiments above (borrowed from our main paper) to prove that without our processing techniques, the synthetic pairs from FreestyleNet cannot benefit the real-image baseline, even downgrading the performance.
> > > >
> > > > We believe our results are convincing. We are eager for your further attention. We appreciate your precious time a lot.
> > > >
> > > > Best regards,
> > > >
> > > > Authors of paper 6586.

---

### Author Rebuttal · Authors · 2023-08-10

**[Contributions]** Our technical contributions mainly lie in three folds:
- **[New target \& new roadmap]** We present a new roadmap to enhance *fully-supervised* semantic segmentation via generating *densely annotated* synthetic images with generative models. Our data-centric perspective is orthogonal to the widely explored model-centric (*e.g.*, network architecture) perspective.
- **[New problem]** We highlight the necessity of designing processing strategies for synthetic images. With our present simple filtering and re-sampling strategies, the model trained with synthetic images can achieve comparable performance with the counterpart of real images, *e.g.*, 48.3 *vs*.  48.5 mIoU on ADE20K and 49.3 *vs*. 50.5 on COCO-Stuff.
- **[Stronger performance]** We achieve 2.0% improvement on average on ADE20K across seven architectures, which is a much larger gap than previous model-centric works achieved (mostly 1%). We believe this will inspire more future works to investigate this promising direction.

**[Uniqueness]** Our work is distinguished from existing works in that:
- **[Few-shot *vs*. fully-supervised]** Existing works only focus on a constrained scenario, *e.g.*, few-shot labels, while we address the challenging but widely acknowledged fully-supervised scenario.
- **[Classification *vs*. semantic segmentation]** Existing works mostly address the classification task, which is cheap to label and even not so urgent for human labels (unsupervised learning performs quite well), while our targeted semantic segmentation task is highly expensive and laborious to annotate.
- **[Image-to-pseudo-mask *vs*. mask-to-image]** Existing works predict pseudo masks for synthetic images, which is not precise enough, while we adopt a mask-to-image synthesis pipeline to produce better aligned image-mask pairs, as evidenced by our response Q2-5 to Reviewer 6xi1.
- **[Blindly using *vs*. carefully processing synthetic data]** Existing works of learning from synthetic data ignore the importance of processing synthetic data discriminatively, while our proposed filtering and re-sampling can impressively improve the effectiveness of synthetic data and ultimately yield a much stronger model.

Lastly, we hope to highlight the filtering strategy only accounts for a (small) portion of our whole work. With this design, we mainly want to emphasize the necessity of processing synthetic images, **which is rarely considered in previous works**. Indeed, we prefer this motivation, compared with concrete instantiations. We think we raise a new problem for future works about *how to better learn from synthetic data*, instead of simply focusing on better synthesis. Besides, we are aware our proposed filtering design is related to existing works (**@3GaN**) in semi-supervised learning, but we adopt similar motivations for totally different scenarios. We will add more discussions in the revised version.

---

> ### Author Response · Authors · 2023-08-17
> **Looking forward to further discussions**
>
> Dear Reviewers 3GaN, FZD9, and VTnr,
>
> We are deeply grateful for your precious time and constructive feedback. For all raised concerns, we have provided necessary quantitative results or detailed explanations. We are greatly motivated that Reviewer 6xi1 has improved the score from 3 to 5 (borderline accept) and  Reviewer jaKB gives a score of 7 (accept).
>
> We are wondering if there are any remaining concerns from you. We are more than willing to provide further clarifications on any aspect. Thank you very much.
>
> Best regards,
>
> Authors of paper 6586.

---

### Decision · Program_Chairs · 2023-09-21

**Decision:**

Accept (poster)

**Comment:**

Reviewers missed a discussion of mixing-based strategies and a comparison to mixing-based strategies and prior GAN-based approaches to generating datasets. Reviewers were concerned about the triviality of the proposed filtering & sampling (in particular vs. online hard example mining (OHEM)), the contribution to the results, and missing comparisons to loss-based alternatives to resampling.

The rebuttal addressed these concerns by
- an additional experiment on mixing-based strategies and promising to add a discussion.
- an additional comparison against a DatasetGAN-based baseline, as requested by [6xi1].
- an ablation study regarding the filtering & sampling vs baseline FreestyleNet.
- a comparison to OHEM

All reviewers acknowledged the rebuttal and found critical concerns addressed.
In light of the responsive reviewers finding their concerns addressed and the experimental results consistently supporting the value of the proposed contributions, the AC recommends accepting the paper.